# Multi-omics analyses reveal rumen microbes and secondary metabolites that are unique to livestock species

Victor O. Omondi,[1,2] Geoffrey O. Bosire,[2] John M. Onyari,[2] Caleb Kibet,[1] Samuel Mwasya,[1] Vanessa N. Onyonyi,[1] Merid N. Getahun[1]

**ABSTRACT** Ruminant livestock, including cattle, sheep, goats, and camels, possess a distinctive digestive system with complex microbiota communities critical for feed conversion and secondary metabolite production, including greenhouse gases. Yet, there is limited knowledge regarding the diversity of rumen microbes and metabolites benefiting livestock physiology, productivity, climate impact, and defense mechanisms across ruminant species. In this study, we utilized metataxonomics and metabolomics data from four evolutionarily distinct livestock species, which had fed on diverse plant materials like grass, shrubs, and acacia trees, to uncover the unique signature microbes and secondary metabolites. We established the presence of a distinctive anaerobic fungus called *Oontomyces* in camels, while cattle exhibited a higher prevalence of unique microbes like *Psychrobacter*, *Anaeromyces*, *Cyllamyces*, and *Orpinomyces*. Goats hosted *Cleistothelebolus*, and *Liebetanzomyces* was unique to sheep. Furthermore, we identified a set of conserved core microbes, including *Prevotella*, *Rickenellaceae*, *Cladosporium*, and *Pecoramyces,* present in all the ruminants, irrespective of host genetics and dietary composition. This underscores their indispensable role in maintaining crucial physiological functions. Regarding secondary metabolites, camel's rumen is rich in organic acids, goat's rumen is rich in alcohols and hydrocarbons, sheep's rumen is rich in indoles, and cattle's rumen is rich in sesquiterpenes. Additionally, linalool propionate and terpinolene were uniquely found in sheep rumen, while valencene was exclusive to cattle. This may suggest the existence of species-specific microbes and metabolites that require host rumen-microbes' environment balance. These results have implications for manipulating the rumen environment to target specific microbes and secondary metabolite networks, thereby enhancing livestock productivity, resilience, reducing susceptibility to vectors, and environmentally preferred livestock husbandry.

**IMPORTANCE** Rumen fermentation, which depends on feed components and rumen microbes, plays a crucial role in feed conversion and the production of various metabolites important for the physiological functions, health, and environmental smartness of ruminant livestock, in addition to providing food for humans. However, given the complexity and variation of the rumen ecosystem and feed of these various livestock species, combined with inter-individual differences between gut microbial communities, how they influence the rumen secondary metabolites remains elusive. Using metagenomics and metabolomics approaches, we show that each livestock species has a signature microbe(s) and secondary metabolites. These findings may contribute toward understanding the rumen ecosystem, microbiome and metabolite networks, which may provide a gateway to manipulating rumen ecosystem pathways toward making livestock production efficient, sustainable, and environmentally friendly.

**KEYWORDS** ruminants, metabolomics, rumen, fermentation, microbiota, metataxonomic, metabolites

Address correspondence to Merid N. Getahun, mgetahun@icipe.org.

The authors declare no conflict of interest.

See the funding table on p. 17.

Livestock are an important part of the ecosystem, especially because they are a major driver in most rural landscapes, diversifying belowground microbes, soil health, function, fertility, and crop productivity. Globally, it is estimated that more than 1.2 billion people are making a living in the livestock sector across the various value chains (1, 2).

Ruminant livestock provide humans with foods, such as milk and meat from non-human-edible plant material, even in arid and semi-arid ecologies, where crop production is not possible due to erratic rainfall and frequent drought, thus the only means to sustainably use such vast land is through sustainable livestock husbandry. Livestock in arid and semi-arid areas are vulnerable to climate shocks, which can be determined by the duration, frequency, and severity of the shocks, as well as the location of the stocks and related assets (3–5). Additionally, livestock's vulnerability to climate change also depends on their sensitivity, which is determined by the breed, feeding system, efficiency, and resilience (5, 6). For instance, the one-humped camel (*Camelus dromedarius*) is the most efficient and resilient animal well adapted to arid and semi-arid ecologies with limited resources, this is recently evidenced as pastoralists shifted from cattle to camel keeping even at higher altitudes (7–10). This can be taken as a climate change adaptation strategy and has the potential to improve livestock climate resilience if the underlying mechanism is understood. However, the underlying mechanisms responsible for the observed variations in resilience between different livestock are not clear.

The rumen, a large fermentation chamber in ruminant livestock, harbors diverse and complex microbial communities that play crucial roles in the digestion and fermentation of feedstuff (11, 12) and the production of diverse metabolites including greenhouse gases (13–16). The relationship between some members of the microbiome and rumen function is well known (13, 17). The role of diet on microbial diversity has been investigated (16, 18, 19). Whereas host genetics have been studied in determining rumen microbes (20–22), most of the studies have been done on a single species and biased towards cattle, and no comparative studies have been reported between diverse ruminant animals that vary both in feeding regime and resilience, which is the focus of this study.

We hypothesize that livestock vary in their rumen microbes and secondary metabolites that have useful traits for livestock resilience and efficiency. The rumen environment hosts the most complex diverse microbial communities consisting of bacteria, fungi, protozoa, etc. Therefore, understanding the diversity and the pivotal role of the rumen microbes and secondary metabolites in digesting fibrous feed, providing nutrients to the host animal, defense, and determining livestock host-environment interaction is key for sustainable animal husbandry. Pertinent global issues of interest include climate resilience, the fight against climate change, and vector-borne diseases through rumen environment manipulation to make livestock part of the solution. Here, using four ruminant livestock that vary in feeding regime, drought resilience, and disease prevalence, we show that each livestock species created a mutual association with signature microbes and secondary metabolites that provide useful ecological traits.

## RESULTS

### Distribution of bacterial and fungal populations in the rumen

To correlate the secondary metabolites with rumen microbes, we performed genomic analysis of the two main rumen domains, bacteria and fungi. The taxonomic analysis of bacterial and fungal populations in the rumens of cattle, sheep, goats, and camels revealed a variation in the dominance of core groups of rumen microbes among the four ruminants (Fig. 1). A total of 1,052 species-level, bacterial amplicon sequence variants (ASVs) were uniquely identified in camels, 949 in cattle, 1,065 in sheep, and 847 in goats, respectively (Fig. 1B). Whereas 113 bacterial ASVs were shared by all the four ruminants, 187 ASVs were shared by both camels and goats, while 208 ASVs were common in cattle and sheep (Fig. 1B). Additionally, among the analyzed bacterial ASVs, 36,363, 47,659,

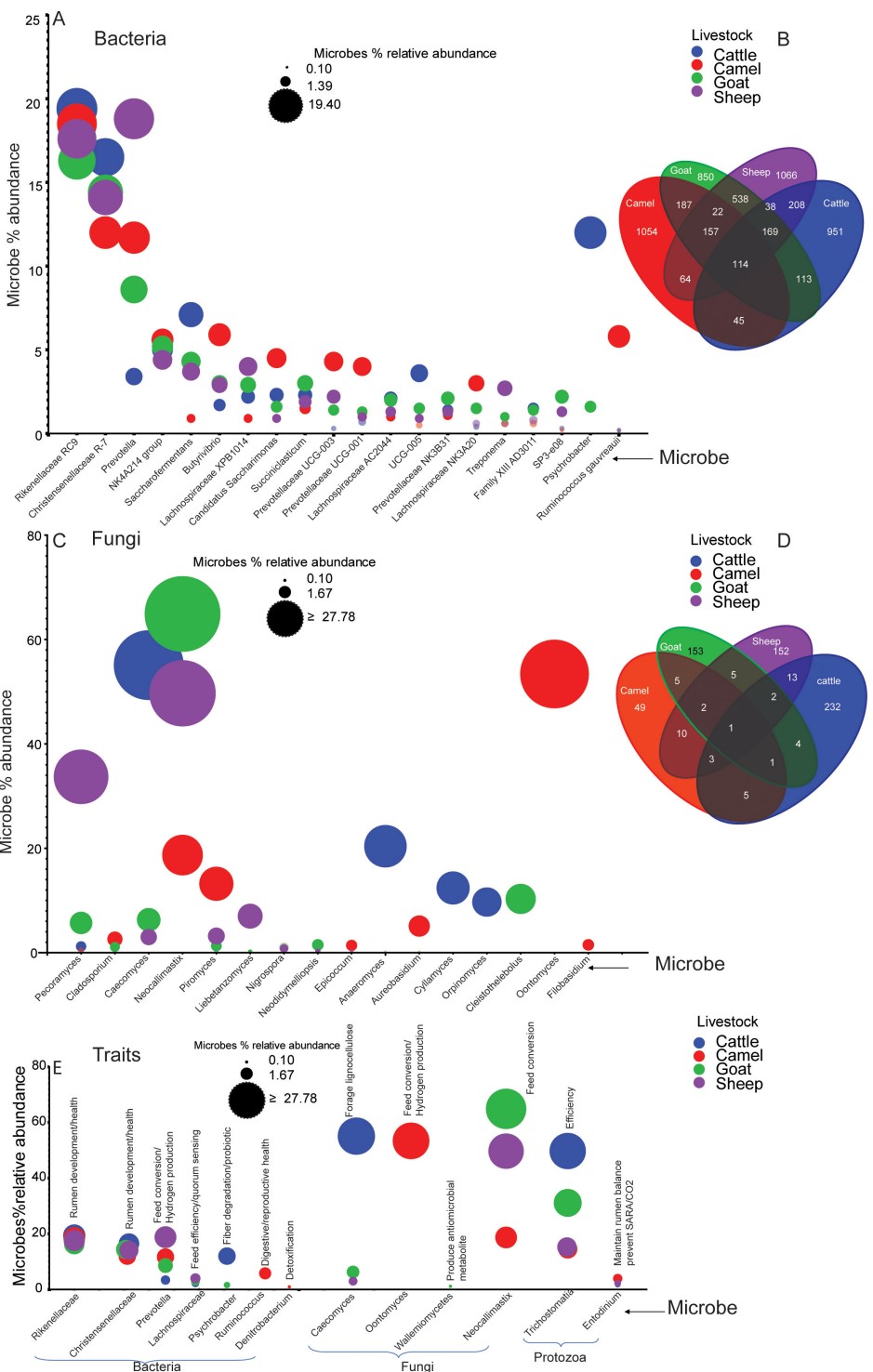

**FIG 1** Dominant bacteria and fungi across the four livestock species. (A) Bubble plots showing the qualitative and quantitative differences of unique and shared bacteria. (B) Venn diagrams showing the total number of the identified unique and shared bacterial microbes. (C) Bubble plots showing the qualitative and quantitative differences of unique and shared fungi communities among the four livestock. (D) Venn diagrams show the total number of the identified unique and shared fungi. (E) Association analysis between selected microbes and host traits. Trait analysis was done based on references 13, 23–38. Plots generated using microbes' relative abundance data with at least 1% relative abundance in one of the four livestock species.

59,338, and 48,806 bacterial ASVs were unclassified in cattle, camels, goats, and sheep, respectively.

Bacteria, being the main members of the rumen microbiome, were widely dominant across the four livestock groups analyzed, comprising most of the species richness, with some bacterial genera being livestock specific (Fig. 1A and B). Further analysis of the identified bacterial ASVs revealed the 20 most abundant bacterial genera present among the four livestock species (Fig. 1A and C). In all four ruminants, the *Rickenellaceae RC9, Christensenellaceae R-7* group*, NK4A214* group, and *Succiniclasticum* group are conserved both in their presence and abundance (Fig. 1A). The genus *Ruminococcus gauvreauii* was abundantly present in camels and in small amount in goats. *Prevotella*, and *Prevotellaceae* a hydrogen-producing bacterial genus, was dominant in camels, however, less abundant in cattle, goats, and sheep (Fig. 1A). The *Psychrobacter* genus was found uniquely in cattle and goats but absent in sheep and camels (Fig. 1A; Table S1). All the remaining bacterial genera were conserved in all four livestock species but with varying abundance. Compared to camels and sheep, cattle and goats had more bacterial diversity due to an additional genus, *Psychrobacter* (Fig. 1A; Table S1).

A comprehensive analysis was conducted on the ASVs of fungi in four livestock species, namely cattle, camels, goats, and sheep. The results showed a total of 232 ASVs in cattle, 49 ASVs in camels, 153 ASVs in goats, and 152 ASVs in sheep at the genus level (Fig. 1C and D). Furthermore, 1,639, 1,450, 6,666, and 1,022 fungal ASVs were unclassified in cattle, camels, goats, and sheep, respectively. Among the identified fungal ASVs, a diverse population of 17 highly prevalent fungal genera was found (Fig. 1C). The analysis further showed that only one fungal ASV was common in all four livestock species, while five were common in camels and goats, whereas cattle and sheep shared 13 ASVs (Fig. 1D). Goats had the highest representation of fungal genera with camels having the least representation among the four ruminants (Fig. 1D). The anaerobic fungal genus *Caecomyces* was abundantly present in cattle and in small amount in goats and sheep but missing in camels. The aerobic fungus genus *Oontomyces* was exclusively found in camels in high abundance. On the other hand, *Neocallimastix* was the most abundant in both goats and sheep, present in small amounts in camels but missing in cattle. *Pecoramyces* was found only in sheep and goats; in the past, it was much more abundant but missing from camels and cattle (Fig. 1C; Table S2). *Liebetanzomyces* were only found in sheep. Furthermore, *Anaeromyces, Orpinomyces,* and *Cyllamyces* were unique to cattle, whereas *Piromyces* were the major groups in camels (Fig. 1C; Table S2). *Nigrospora* was absent in cattle but present in the other three livestock in small amounts. *Caecomyces,* which was dominant in cattle, was absent in camels, though present in both goats and sheep. *Cleistothelebolus* was distinct to goats and may be considered a signature fungal community in goat rumen (Fig. 1C; Table S2). Only *Cladosporium and Pecoramyces* were found to be conserved among the four livestock, thus suggestive of their roles for conserved function. Furthermore, camels harbor different protozoans as compared to other livestock (data not shown).

## Alpha and beta diversity

The four ruminants showed greater diversity in bacterial microbes as revealed by the high Shannon alpha diversity index, which takes into consideration both abundance and richness; however, there was no significant difference between the four livestock species, $F = 1.18$, $P = 0.35$ and Chao1 value $F = 1.59$, $P = 0.23$ (Fig. 2A and B). Beta diversity, variation of bacterial communities between the four livestock species phylogenetically, was assessed by calculating the principal coordinate analysis (PCoA) of different rumen bacterial domains using unweighted UniFrac distance dissimilarity and permutational multivariate analysis of variance (PERMANOVA) permutations 9,999 ($P < 0.05$) based on the Bray-Curtis dissimilarity distance matrix using all ASVs as an input. The investigation indicated that cattle exhibit a high degree of similarity among themselves, whereas camels also demonstrate a comparable level of similarity within themselves. Neverthe-

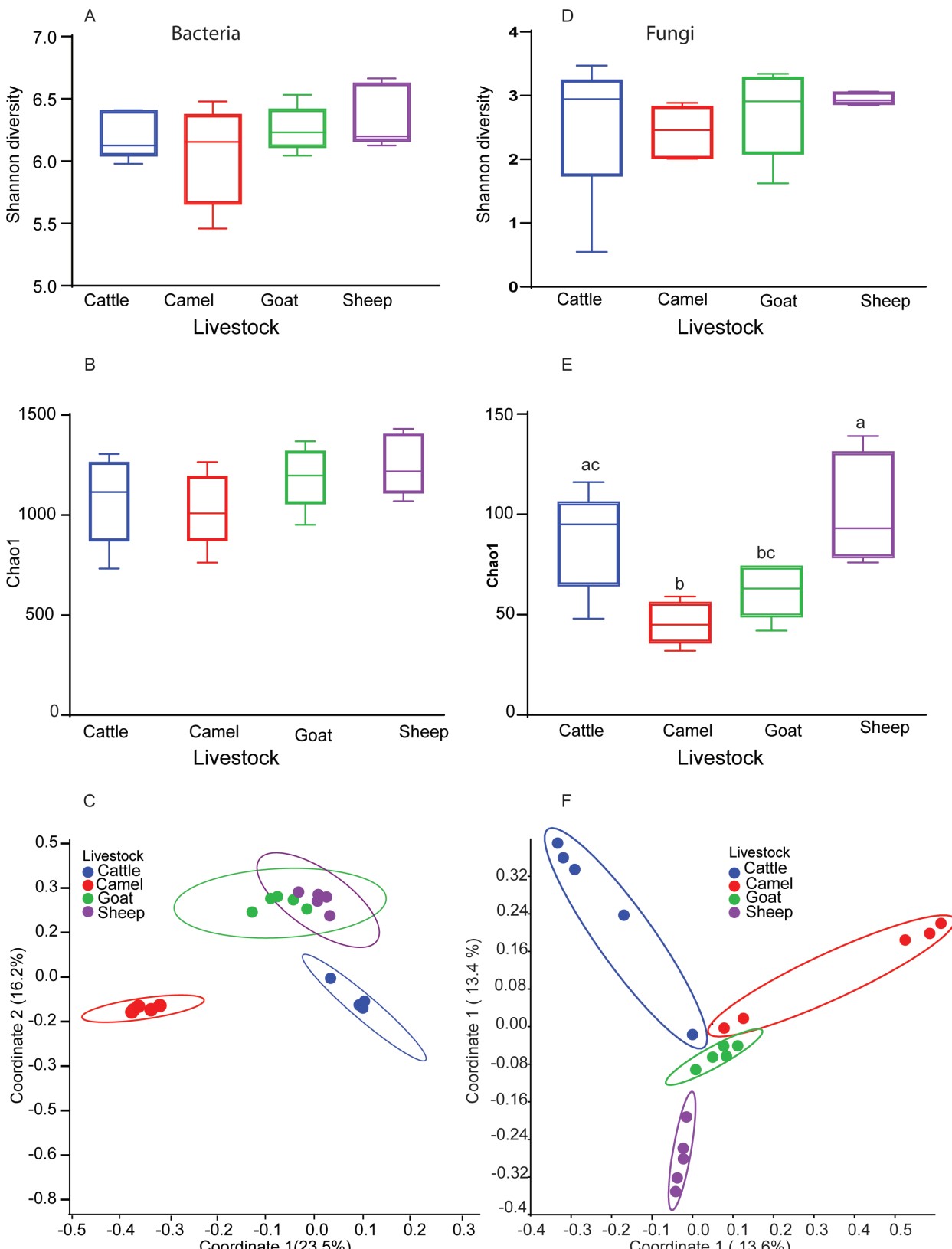

**FIG 2** Diversity of bacteria and fungi across the four livestock species. (A) Shannon diversity index for bacteria. (B) Chao1 index for bacteria. (C) Beta diversity PCoA ellipse clusters showing the distribution of bacteria. (D) Shannon diversity index for fungi. (E) Chao1 richness index for fungi. (F) Beta diversity PCoA ellipse clusters showing the distribution of fungi.

less, the goats and sheep exhibited a close grouping, indicating a lack of distinction between the two species (Fig. 2C).

Furthermore, the Shannon diversity index (Fig. 2D) did not show any notable variations in fungal microorganisms across the four livestock species. Nevertheless, the Chao1 index, which specifically quantifies microbial richness, indicated that camels had lower richness compared to cattle and sheep ($F = 7.63$, $P = 0.002$, analysis of variance [ANOVA] followed by Tukey's multiple comparisons test) (Fig. 2E). The beta diversity analysis showed that there is individual variation between cattle and camel populations but still tend to cluster separately ($P < 0.05$, PERMANOVA) (Fig. 2F).

## Dietary composition assessment in livestock rumen

Besides the host's individual genetic makeup, the composition of the host's diet influences the types and amounts of substrates available to the microbial community, which, in turn, influence the production of secondary metabolites (20). Therefore, we evaluated the dietary composition in the rumens of the four livestock species. We established that the four ruminants fed on diverse plant materials. This may be attributed to the fact that feeding in pastoralist setup is not controlled or restricted, and hence livestock have access to a wide range of plant materials. For instance, we found that in addition to grasses (*Poaceae*), *Cenchus cilliaris* and *Cenchus americanus,* which had been consumed by cattle and sheep, cattle had consumed other plant species such as *Rhus gueinzii and Rhus transvaalensii* despite being predominantly grazers (Fig. 3). Unlike cattle and sheep, camels and goats are specially adapted to feed on leaves, fruits of high-growing woody plants, soft shoots, and shrubs, such as *Acacia concinna*, *Paraprenanthes sororia*, *Vachellia nilotica,* and *Searsia tripartita* (Fig. 3), which are predominantly found within arid and semi-arid areas. Therefore, it points to the diversity in dietary composition among the ruminants, which influences both the metabolite compound and microbial population composition among the ruminants.

## Ruminal metabolite composition in livestock

In the present study, a total of 162 metabolite compounds (Table S3) were identified in the bovine rumen content of four livestock; cattle, sheep, goats, and camels by gas chromatography-mass spectrometry (GC-MS). The detected compounds represented various chemical classes, including alcohols, ketones, phenols, volatile fatty acids, terpenes, esters, and hydrocarbons. Although most major classes of secondary metabolites have ubiquitous distributions among the four livestock, each livestock species has its own signature secondary metabolites (Fig. 4A). A random forest classification was conducted to reveal the top 10 predictive compounds for individual ruminant species (Fig. 4B). 2, 6 dimethyl 4-octene, 3-methylbutanoic acid, 1, 3-cyclohexene, and tricyclene being the most predictive secondary metabolite compounds of cattle, camels, goats, and sheep rumen, respectively. In camel, 5 out of the 10 predictive compounds are acids, signifying the diversity of acid in camel rumen. However, we did not observe the dominance of any specific chemical class in the other livestock; diverse classes of compounds contribute to predictive signature odors. The diversity and contrast in metabolite composition among the ruminants were revealed by the clustering and segregation of the respective species based on their metabolite composition by multidimensional scaling (MDS) and matrix plot (Fig. 5A and B). While some species, such as cattle and sheep, which are grazers clustered in proximity, camels and goats, which are browsers, were distinctly clustered apart from the other ruminants and from each other (Fig. 5B), thus, demonstrating a similarity in metabolite composition among grazers (cattle and sheep) but not clear with browsers (camels and goats). Overall, the four livestock dissimilarity based on their rumen secondary metabolites was 72.5%. Twenty-three compounds, based on quantitative and qualitative differences contributed to more than 50% of the variation (Fig. 5C). Specific metabolite compounds were found to be unique to specific livestock and some were shared between two or more ruminant species. For instance, compounds like isoamyl benzyl ether and beta-citronellene were

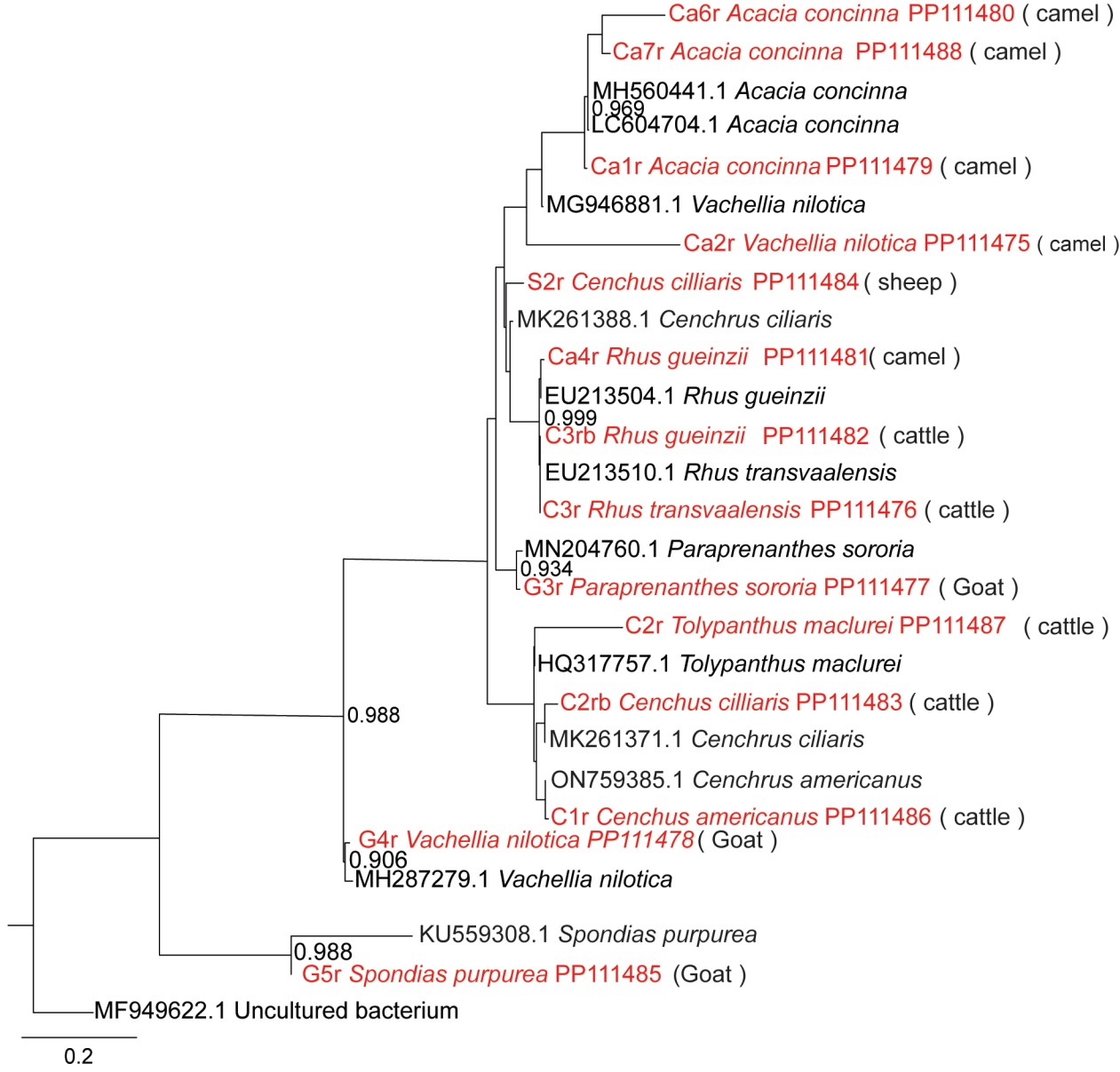

**FIG 3** Phylogenetic tree showing plant diet composition in the rumens of the four livestock. The plant species identified in this study from each livestock and their corressponding accession numbers are indicated in red, while their ex-type strains are shown in black. Phylogenetic analysis based on maximum likelihood with a bootstrap of 1,000 pseudoreplicates, with MF949622.1 uncultured bacterium as the outgroup taxa. Maximum likelihood bootstrap values equal to or greater than 65% are shown on the nodes, while the scale bar indicates 0.2 changes.

absent in cattle, while linalool propionate and Valencene were distinctively detected in sheep and cattle, respectively. Similarly, cis-calamine and 1, 5, 9-undecatriene were absent among camels, while β-gurjunene was absent in both camels and goats and limonene was missing in cattle. Additionally, 1-p menthene was not found in cattle. On the other hand, 1, 3-cyclohexadiene is found solely in goat and sheep rumens.

## Metabolite composition by chemical functional groups

We then evaluated the chemical identities and variation in the distribution of volatile organic compounds across the four ruminant species, with compounds categorized based on their functional group classification, including phenols, alcohols, indoles, monoterpenes, sesquiterpenes, acids, hydrocarbons, and ketones. We found significant differences in the relative abundance of certain chemical classes, such as alcohols,

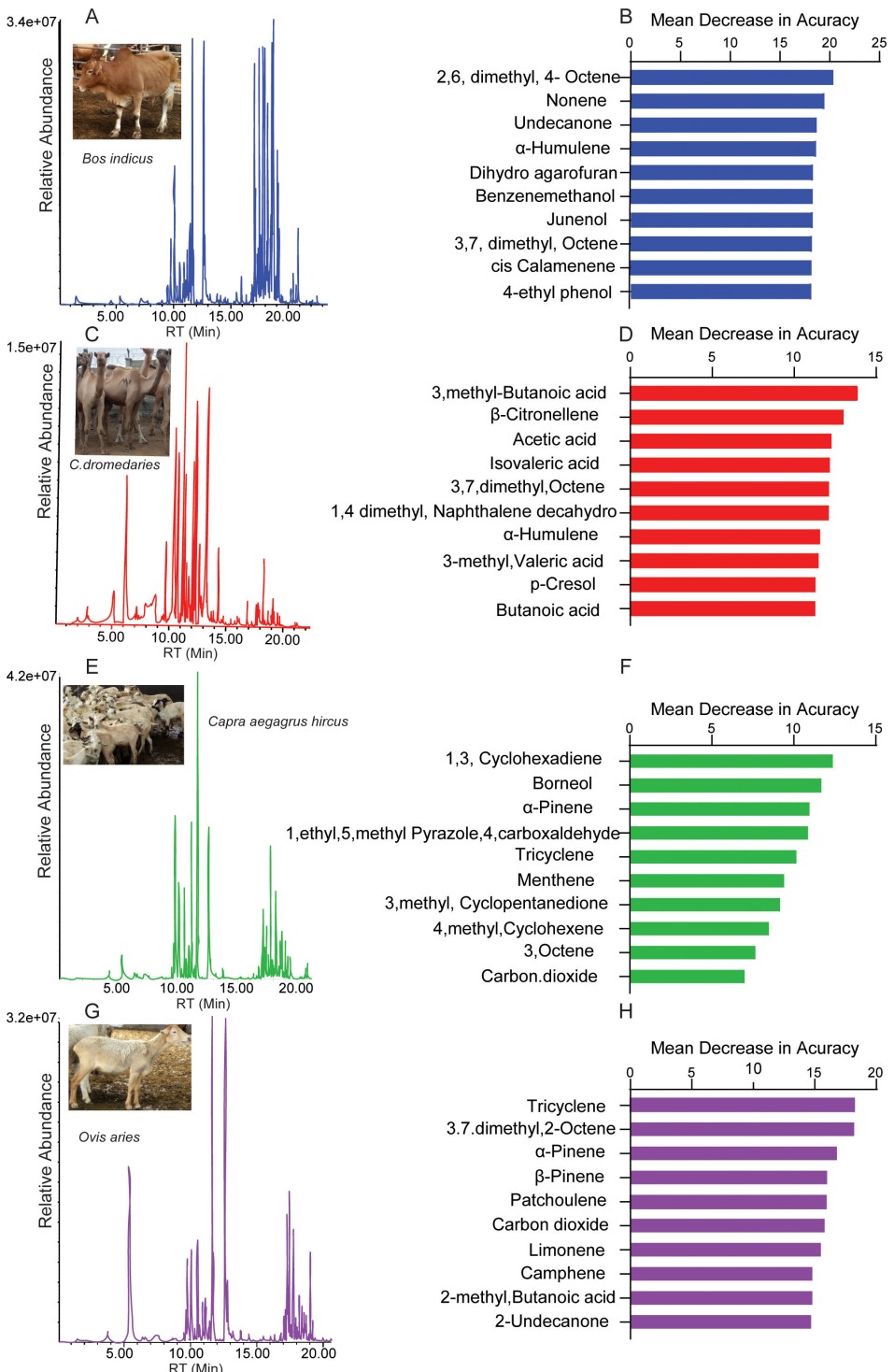

**FIG 4** Rumen secondary metabolite profiles for various ruminant livestock. (A, C, E, and G) GC-MS chromatogram of metabolite compounds in the ruminal fluid of various livestock, cattle, camels, sheep, and goats, respectively. ( B, D, F, H) Histograms showing the classification of the top 10 important compounds from different livestock rumens based on their mean decrease in accuracy (MDA) of the random forest analysis. Metabolites with the highest MDA value as shown on the histogram are the most important and consequently most predictive for each species. Animals' pictures are original.

hydrocarbons, monoterpenes, acids, and sesquiterpenes, among the four livestock groups, with cattle, sheep, camels, and goats displaying varying relative abundance of these compounds (Fig. 6A through G, $P < 0.05$). Cattle, sheep, and camels had

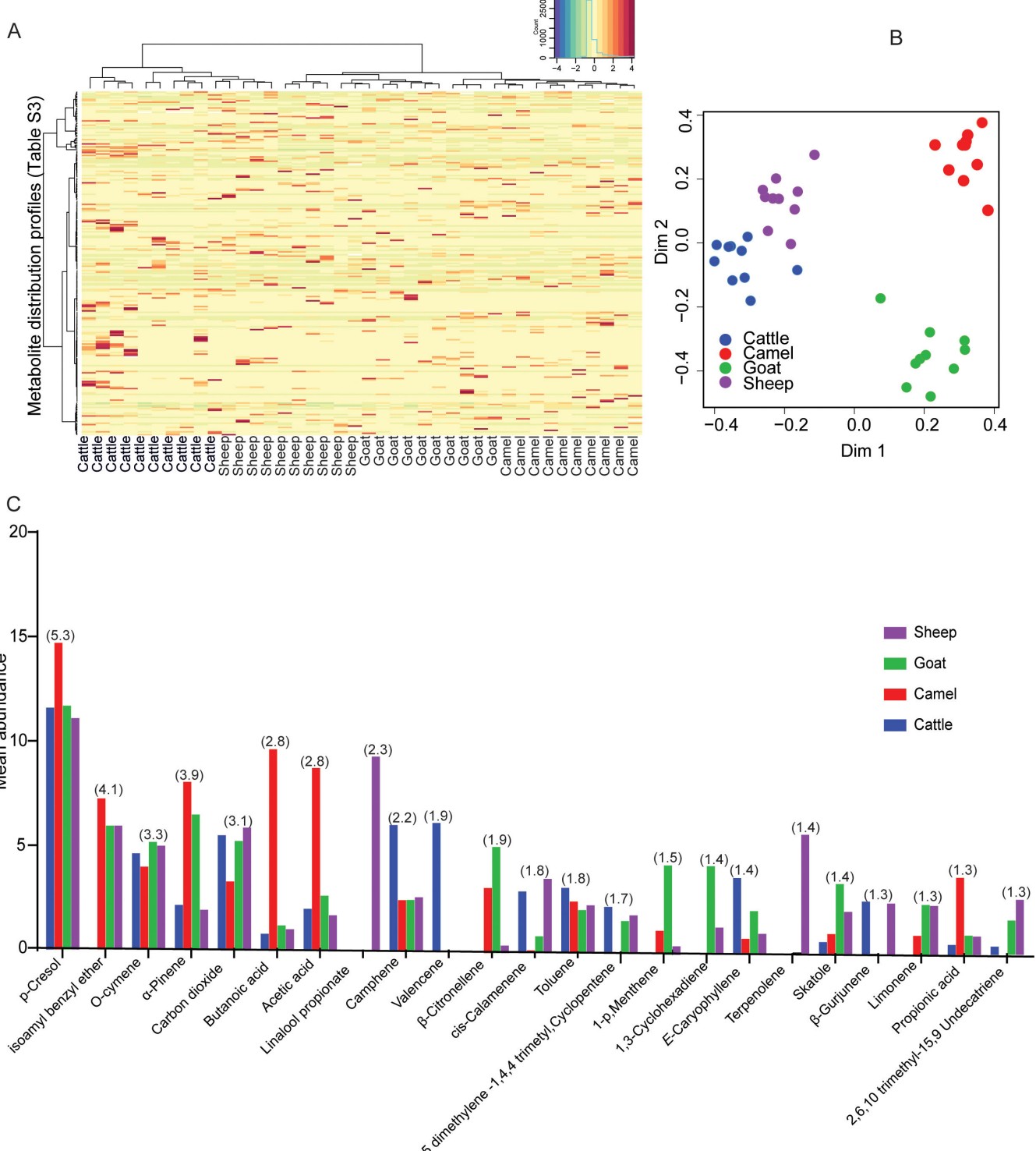

FIG 5  Diversity of rumen secondary metabolites across the four livestock species. (A) Heatmap-coded matrix showing the relative percent contribution of individual compounds to the total composition of each livestock species (for details, please see Table S3). (B) MDS plot showing the segregation of ruminants based on metabolite composition. (C) Histogram showing the classification of the top 23 metabolite compounds contributed to 50% dissimilarity between the four livestock based on similarity percentage analysis; the number in parenthesis is the percent contribution of a given compound for the dissimilarity.

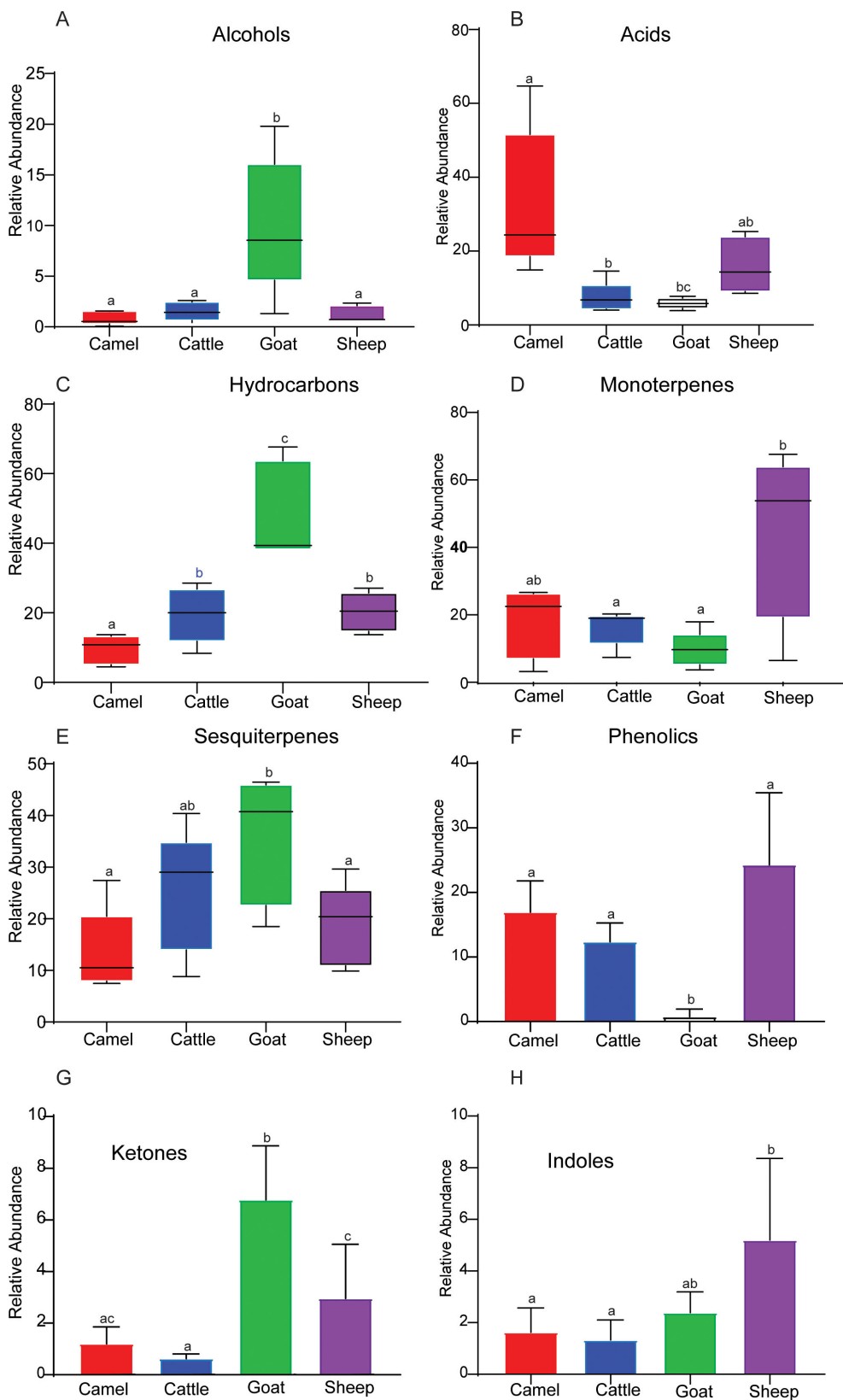

**FIG 6** Dominant secondary metabolites by functional groups. (A–H) Box plots showing variations of chemical families of identified metabolite compounds among different ruminants. Bar graphs followed by different letters are statistically different in their abundance based on the analysis of variance ($P \leq 0.05$).

significantly lower alcohol and sesquiterpene concentrations compared to goats (Fig. 6A and E, ANOVA, $P < 0.05$). Similarly, camels were established to contain the highest acid concentration compared to cattle, sheep, and goats (ANOVA, $P = 0.004$, Fig. 6B), which resulted in an acidic rumen environment in camels that had been established when the rumen pH was determined for the four ruminants. Camel ruminal pH was significantly lower (pH 6.3–6.5) compared to cattle, goats, and sheep, which had a relatively neutral pH ranging from 7.0 to 7.4 (ANOVA, $P < 0.05$). Interestingly, when we examined the carbon dioxide emissions, the three livestock species (cattle, sheep, and goats) produced almost twice as much carbon dioxide as camels. This intriguing array of differences in chemical composition and carbon dioxide emissions among livestock highlights the diversity of these animals in our environment. On the other hand, phenols, ketones, and hydrocarbons were more abundant in goats compared to other livestock, and sheep had more indoles compared to other livestock (ANOVA, $P < 0.05$, Fig. 6H).

## DISCUSSION

### Microbiome and diet composition

In this comparative study, we reveal an intricate network of rumen microorganisms that coexist, interact, and vie for resources, ultimately maintaining a delicate equilibrium of diverse end products, including secondary metabolites. These products not only fuel microbial growth without autotoxicity but also yield beneficial outcomes for the host. A better understanding of the impact of the rumen microbiome on host well-being and performance could open the door to innovative approaches and strategies for enhancing desired traits in livestock using nature-inspired methods.

Three bacterial genera (*Rikenellaceae* RC9 gut group*, Prevotella,* NK4A214 group, and *Christensenellaceae* R-7 group) were highly conserved both in their presence and abundance among the four livestock species regardless of genetics. This may suggest that they are core rumen bacteria, essential for highly conserved common traits or functions for these animals (Fig. 1E). However, there were unique bacteria in each host. For instance, the detection of *Anaeromyces, Cyllamyces*, and *Orpinomyces* in cattle only may be suggestive of the existence of species-specific microbes that require host rumen-microbes' environment balance. Similarly, two fungi genera *Cladosporium* and *Pecoramyces* are conserved among the four livestock, although varied in abundance. This demonstrates that they are essential for a conserved function, which may be dependent on their relative abundance in each species.

Several fungi genera were unique. Some were present only in one animal species, while some others were shared between either two or three animals but not by all four. Even those that were shared varied in their abundance, which may be integral to their environment and potentially compatible with the rumen environment and host requirements. The significant variation in fungi microbes between the four ruminant livestock may be explained to a significant extent by host genetics (13, 39–41). This unique microbe-host framework variation in microbial composition between hosts may affect microbially mediated ecosystem processes. Additionally, this variation may as well be dependent on host phylogenetic relatedness and trait-based patterns of ecologies (39).

Each of the bacterial and fungal communities established in the present study plays a specific metabolic role in the rumen (42–44). For instance, bacterial species like *Ruminococcus, Lachnospiraceae, Christensenellaceae,* and *Prevotella* are associated with hydrogen production during rumen fermentation (22, 44). There are some microbes that were ubiquitous in all four ruminant species, demonstrating their wide rumen environment adaptation. For instance, camel rumen has an acidic pH compared to other livestock.

The next step is to explicitly link the observed microbial and secondary metabolite diversity and network with the basic evolutionary principle, which is biological fitness. We have shown the variation between diverse microbes among the four livestock that varies in feeding behavior, drought resilience, and disease susceptibility (45). For

instance, camels and small ruminants, to some extent, are the most resilient to frequent drought among the analyzed livestock compared to cattle (7–10). This could be due to the abundant presence of unique anaerobic fungi, *Oontomyces*, originally identified from Indian camel, (46) and bacteria (*Prevotella*) in camels that have demonstrated high capability of diet conversion (23, 24, 39, 47). Additionally, the fungus *Neocallimastix,* which is present in camels, sheep, and goats but absent in cattle, has been shown to be effective in the bioconversion of poor diet such as lignocellulose into useful products (25, 48), which may have contributed to their resilience. These microbes combined with other physiological mechanisms such as suppression of cholesterol biosynthesis in the kidneys of camels and retention and reabsorption of water (49) may be responsible for camels' drought resilience. Thus, camel's evolutionary success to dry climate is partly may be due to the ability to engage in mutualistic interactions with useful microbes that provide novel ecological adaptation traits. Furthermore, such knowledge will give us the opportunity to manipulate the rumen environment to make livestock less susceptible to vectors, efficient in converting their diet to animal protein, and to make livestock environmentally friendly. For instance, in one study, the addition of a fungal inoculant to the diet of dairy cows was found to increase the production of propionate and decrease the production of acetate (50), which is a precursor of greenhouse gas production. Furthermore, microbiome work in humans and rodents has revealed that microbes play essential roles in host health and function (51, 52). Similarly, in our previous work, these various livestock exhibited various susceptibilities to various pathogens (53), which may depend on their mutualistic association with useful microbes.

A recent study to establish the role of the rumen microbiome in dairy cow productivity and greenhouse gas emissions revealed that a heritable subset of the core rumen microbiome influenced the efficiency of feed utilization and the environmental impact of dairy farming (13). This demonstrated that the core microbiome had a significant explanatory role in relation to dietary components within a controlled experimental setting. In our experiment, it was difficult to dissect the role of diet for the microbes and chemo-diversity as the animals were from a free grazing and browsing setup and fed on diverse diets. If we assume that diet may structure rumen microbes, we would have expected similarity both in microbes and secondary metabolites between browsers (camels and goats) and between grazers (cattle and sheep). However, we did not establish a clear link between diet and microbes. For instance, only one bacterial genus, *Psychrobacter,* was absent in camels and sheep. If diet shapes the rumen microbes, browsers (camels and goats) should share more similar microbes than camels have in common with cattle and sheep, and vice versa. On the other hand, if microbes dictate diet, cattle and sheep share more similar microbes than what camels and cattle share. But sheep and goats shared more bacteria than either of them shared between camel and cattle. This may also be because there are no strict browsers and grazers under the free grazing setting, as they can easily shift between various diets depending on feed availability.

The various plants consumed by the various livestock are characterized by high fiber content, rich in secondary metabolites, and bioactive compounds including tannins, flavonoids, alkaloids, and terpenoids, hence may have potential health benefits for ruminants (54–56). The utilization of shrubs and woody plants in livestock diets has been shown to increase rumen metabolite richness compared to diets based on traditional forage sources (57–59). Studies showed that feeding goats on *Acacia saligna*, a shrub species, led to increased diversity and richness of rumen metabolites compared to a control diet based on alfalfa hay (60–63). The composition of the plant diet can have significant impacts on the production of metabolites in the rumen. For instance, Grasses (*Poaceae*) contain fermentable cellulose, hemicellulose, lignin, and protein, which are broken down by rumen microbes into various metabolites, including acetate and propionate (17, 64), which are ingredients in greenhouse gas formation and energy source. Hence, the variability in plant diets can have a significant impact on rumen metabolite production and composition in livestock.

## Secondary metabolite composition and diversity

Rumen fermentation is a complex process that results in the production of various metabolites (13, 50). We established a wide range of secondary metabolite compounds in the rumens of the four livestock species. This highlights the interplay between host genetics, diet, and microbes, most of which are associated with various biochemical activities in livestock rumen. The detection of metabolite compound classes, such as volatile fatty acids, aromatic hydrocarbons, terpenes, hydrocarbons, phenols, and alcohols, displays the diversity and complexity of metabolic synthetic pathways in livestock rumens, leading to the production of several diverse metabolites (63). The detection of plant-derived metabolite compounds, such as camphene, α-pinene, and β-caryophyllene, including fecal predictive indolic and phenolic compounds like p-cresol (a byproduct of protein breakdown in the animal gut) and skatole, which had previously been reported in various animals metabolic by-products like animal feces, has a role in livestock-vectors interaction (65–68). This demonstrates that metabolites are conserved as they pass through various digestion processes. But we also observed less complexity in some metabolites. For example, phenols in the rumen are less complex compared to livestock urine (53), which shows that metabolites may gain complexity after they leave the rumen.

Even though the examined metabolite composition varied among the ruminants, minimal intraspecific variation was realized among individual species herd (Fig. S1). Despite the intraspecific diversity among the four ruminants, identical metabolite compound classes were detected, suggesting shared biosynthetic pathways during rumen metabolism. This phenomenon further points to the possibility of some metabolite compounds having a conserved function regardless of the host genotype. This has also been shown by some studies, which also documented that rumen secondary metabolites may not be affected by host-specific microbes, host genotype, or livestock population dynamics (13, 16, 26, 69).

Studies have highlighted a direct relationship between bacterial and fungal populace with rumen metabolome (70–72). These microorganisms work together in a symbiotic relationship with the host to break down complex plant polysaccharides and fiber into simple sugars, which can then be fermented into volatile fatty acids, microbial proteins, and other metabolites that can be absorbed by the host animal (16, 73). Thus, the various secondary metabolites identified may provide various functions to the host. Ruminants, such as cattle, sheep, and goats, utilize hydrocarbons as an energy source, largely contained in plant carbohydrates like glucose and sucrose, by fermenting them in their rumen into volatile fatty acids, which are then absorbed and utilized for energy (74, 75). In this study, we established notable differences in hydrocarbons, terpenes, ketones, and indoles relative abundance among the ruminants. Such variations clarify relevant aspects, such as diet composition, breed, and environment, since the detection and concentration of most ruminal metabolite compounds are influenced by these factors (75–78). The diversity and importance of different compound classes of rumen metabolome in livestock were further demonstrated by the detection of terpenes, which have been linked to improved nutrient utilization and digestive health (79). In addition to terpenes, chemical compound classes like acids, phenols, indoles, ketones, and alcohols also varied significantly among the ruminants (Fig. 6).

The acid profile significantly differs between the four livestock species, with camels having the highest. Acids are involved in the hydrolysis of complex carbohydrates, such as cellulose, lignin, and hemicellulose, into simpler sugars that can be further metabolized by rumen microbes (80). This may be ascribed to the fact that acids are energy sources for the host animal and can be used as precursors for energy production during special conditions. For instance, fatty acids, such as acetate, are used by the host animal as a precursor for fatty acid synthesis in adipose tissues, which can then be utilized as an energy source during times of high energy demand, such as during lactation or periods of feed restriction (80); thus, the diverse acids produced in camel rumen may have contributed to camel rumen acidic pH and resilience even during extended

droughts with limited feed availability in arid and semi-arid ecologies. Phenols and indoles are aromatic compounds that are derived from lignin, which is present in the cell wall of plants in addition to being produced during the fermentation of plant material in the rumen. Phenols and ketones exhibit antimicrobial properties that can help to maintain a healthy microbial balance in the rumen (81). Additionally, the antioxidant properties of both ketones and phenols can help reduce oxidative stress in the rumen and improve animal health (82, 83). Alcohols provide energy for rumen microbes in addition to being a carbon source for the synthesis of microbial protein (84, 85). On the other hand, ketones have been shown to be an alternative energy source for ruminants in addition to preventing ketosis (86). Furthermore, elucidation of maternal, genetic, and environmental factors, such as rumen environment (for instance, pH, nutrient, etc.), may provide novel insights into possible mechanisms for manipulating the rumen microbial and secondary metabolites composition to enhance long-term host health, performance, and climate resilience.

## Conclusion

This study sheds light on the intricate web of interactions within the rumen ecosystem, highlighting the coexistence, interaction, and competition among various microorganisms. This interplay results in a delicate balance of end products, including secondary metabolites that fuel microbial growth and offer benefits to the host animals. The identification of highly conserved microbial genera across different livestock species suggests the existence of core rumen microbes essential for common traits or functions. Simultaneously, the presence of unique bacteria and fungi in specific hosts underscores the role of species-specific microbes in maintaining the balance of the rumen environment. Furthermore, this study also highlights the intricate relationship between diet, host genetics, and microbial composition in the rumen, which contributes to the diversity of secondary metabolites. These metabolites play critical roles in livestock rumen metabolism. While certain compound classes are conserved among different ruminants, their abundance can vary significantly, which can, in turn, confer unique ecological traits to the host organisms.

Our results demonstrate that rumen fermentation at the interface of host genetics, microbes, and diets has a significant implication in the production of complex secondary metabolites. Hence, it has the potential to revolutionize the livestock sector. By linking the rumen microbiome to host health and performance, we can explore novel strategies and treatments inspired by nature. For instance, the microbial communities in camels and goats may contribute to their resilience in arid environments, offering valuable insights into drought-resistant livestock breeding. Furthermore, the manipulation of the rumen environment could lead to livestock that are less susceptible to disease vectors and more efficient at converting their diet into animal protein while reducing greenhouse gas production. Overall, this study underscores the importance of the rumen microbiome and its associated secondary metabolites in shaping the health, performance, resilience, and environmental impact of livestock. By further exploring the evolutionary and ecological aspects of these interactions, we can unlock new opportunities for sustainable and environmentally friendly livestock husbandry.

## MATERIALS AND METHODS

### Collection of rumen content

Bovine rumen contents were collected from 10 different freshly slaughtered Boran cattle (*Bos indicus*), goats (*Capra aegagrus hircus*), sheep (*Ovis aries*), and camels (*Camelus dromedaries*) from their respective slaughterhouses in Nairobi (−1.18623, 36.90744) and Machakos (−1.46500, 36.98166) Counties in Kenya, Africa. The samples (500 mL each) were kept in sterile airtight freeze-resistant 1 L odor collection glass jars (Sigma Scientific,

USA) and transported in a cooler box to the laboratory for metabolite compound collection and analysis.

## Genomic DNA extraction

To extract genomic DNA from the rumen contents of cattle, sheep, camels, and goats, 200 µL of the sample was mixed with an equal volume of buffered phenol and 20 µL of 20% SDS in a 2 mL centrifuge tube (Eppendorf, Germany). After adding 0.5 g of 2 mm zirconia beads (BioSpec Inc., USA), the mixture was shaken thrice in a mini tissue lyser (Qiagen, Hilden, Germany) at a frequency of 30 Hz for 90 seconds. The lysate was then centrifuged at 14,000 rpm on a 5417R centrifuge (Eppendorf, Germany) for 10 minutes, and the supernatant was transferred to a 1.5 mL clean tube (Eppendorf, Germany). Afterward, 200 µL of buffered phenol was added to the supernatant, the mixture was briefly vortexed, and then centrifuged at 14,000 rpm at 4°C for 15 minutes. The DNA was then precipitated by adding 500 µL absolute ethanol to the supernatant in a clean 1.5 mL centrifuge tube and centrifuged at 14,000 rpm at 4°C for 5 minutes. The supernatant was discarded, and the DNA pellet was washed with 500 µL of 70% ethanol and then centrifuged for 5 minutes. Finally, the pellet was suspended in 100 µL of preheated elution buffer G (ISOLATE II Genomic DNA kit, Bioline Meridian). The DNA quality and quantity were checked by a Nanodrop spectrophotometer (Thermo Scientific, Wilmington, DE, USA). Aliquots of 50 µL of the obtained DNA extracts were sent to Macrogen Inc. (Netherlands) for Illumina next-generation sequencing targeting 16S rRNA and ITS1 for bacteria and fungi, respectively. The remaining amounts (50 µL) were utilized for PCR for plant diet identification.

## PCR amplification for diet composition screening

PCR amplification targeting two chloroplast markers, consisting of coding (rbcL gene) and non-coding gene spacer region (trnH-psbA) primers (Table S4), was done according to reference (87). The obtained amplicons were then sent to Macrogen Inc (Netherlands) for sequencing. Using Geneious software, obtained sequences were cleaned, edited, and aligned, resulting in a congruent sequence made up of contigs from both the forward and reverse sequences. The plant species were then identified by aligning the processed sequences against the GenBank database using the NCBI BLAST1 search engine. Subsequent phylogenetic analyses were done using the MEGA software version 11 (88).

## Metabolite extraction and analysis

Metabolite compounds from cattle, camel, sheep, and goat rumen contents were extracted using the headspace, solid phase microextraction (HS-SPME) technique as detailed in reference (89). Stableflex 24Ga, manual holder SPME fibers (65 µm, polydimethyl siloxane/divinylbezene, Supelco, Bellefonte, PA, USA) were used to trap the volatile metabolite compounds and later analyzed by gas chromatography (HP-7890A, Agilent Technologies, USA) coupled with mass spectrometry (5975C, Agilent Technologies, USA). The chromatograms were subsequently examined using the Agilent MSD Productivity ChemStation software designed for GC and GC/MS systems (Agilent Technologies, USA). The integration process for the respective compound peaks employed a probability-based matching algorithm, with an initial peak width set at 0.034 and an initial threshold of 15.7. To identify individual compounds, a computer-aided approach was used, comparing their retention times and corresponding mass spectral data to the MSD library (specifically, NIST 2005, NIST 05a, and Adams MS HP, all from the USA). For a compound to be considered correctly identified, its spectra needed to exhibit a minimum probability match factor exceeding 80%. Metabolite compounds that were present in at least 7 out of the 10 ruminal fluid samples analyzed for each livestock species were considered as positively detected in the rumen (89).

## Data analysis

Multivariate statistical analyses were conducted based on the nature of the obtained data using R studio statistical software version 4.2.1 (90), PAST software Version 4.03 (91), and GraphPad Prism version 9. Similarity percentages (SIMPER) and one-way ANOSIM with Bray-Curtis dissimilarity index were used to compare the profiles and establish the dissimilarity contribution of identified metabolite compounds based on their peak areas across the four livestock species. The metabolite compounds were then classified using the R software package "Random Forest," version 4.2.1. The random forest analysis was executed by running 1,000 iterations (ntree) with 10 compounds randomly selected at each split (mtry = $\sqrt{q}$, where $q$ is the total number of compounds). Based on the function "importance ()," we generated the mean decrease in accuracy (MDA), which provides an important score for each metabolite compound. For each livestock, the metabolite with the highest MDA value was considered the most important. An MDS plot and a classical cluster dendrogram were used to visualize the output of analyzed metabolite compound profiles in each livestock. We then used Pearson's correlation to establish how metabolite compounds compared among individual ruminants' herd populations. The detected metabolite compounds from across the four ruminants were then pooled based on their chemical identities, after checking for normality using the Shapiro-Wilk test ($P > 0.05$). A pairwise comparison of the mean relative abundance of respective metabolite compounds in each chemical entity was analyzed by ANOVA among the four ruminants. Statistical significance was declared at $P < 0.05$.

## Bioinformatics analysis

Initially, the data obtained from Illumina sequencing was assessed using nf-core-ampliseq (v2.4.0) workflow and nextflow (v22.10.0), with predefined parameters of trunclenf = 180 and trunclenr = 120. The workflow proceeded as follows: first, the quality of the reads was checked, using FASTQC (version 0.11.9). Cutadapt (v4.1) was then employed to trim reads and eliminate adapter sequences, following the method developed by Martin (92). Preprocessing was performed using the DADA2 tool (v1.26.0) for filtering and trimming, dereplication, sample inference, merging of paired-end reads, removal of chimeras, and taxonomic classification of the ASVs, as outlined in reference (93). Furthermore, DADA2 performed the classification of the ASVs' taxa based on their taxonomic categorization (Silva database v138 was used on 16S rRNA, and unite database v8.3 was used on ITS1 rRNA).

## Abundance visualization

To visualize the ASV count table and ASV taxonomy table generated by the DADA2 algorithm within the nf-core ampliseq workflow, R statistical software (version 4.2.1) was used for further analysis. The ASV count table, ASV taxonomy table, and metadata were put into a single phyloseq object using the Phyloseq package (version 1.40.0) in R. A subset_taxa() function was then employed to eliminate undesired taxa before converting it to a data frame for further manipulation using the phyloseq_to_df() function. Subsequent data frame manipulation was conducted by tidyverse package (version 1.3.2). Finally, ggplot2 (version 3.4.0) and Cairo (version 1.6.0) were used to produce the visual plots.

## Alpha and beta diversity

The reads were rarefied to uniform sequencing depths using microbiome package (version 1.18.0), and rarefaction curves were drawn with vegan (version 2.6.2) using function rarecurves(). MicrobiotaProcess (v1.8.2) and ggplot2 (v3.3.6) were used to visualize alpha and beta diversity. Three alpha diversity metrics, the Shannon, Chao1, and Evenness indices, were used. We used the Shapiro-Wilk normality test followed by Kruskal-Wallis to determine the statistical significance of the alpha diversity metrics.

The PCoA of beta diversity was determined on weighted UniFrac distance matrix with plot_ordination() function, and PERMANOVA permutations 999 ($P < 0.05$) were performed based on the Bray-Curtis dissimilarity distance matrix.

## ACKNOWLEDGMENTS

The authors gratefully acknowledge the financial support for this research by the following organizations and agencies: Max Planck Institute icipe partner group to M.N.G. and the Swedish International Development Cooperation Agency (Sida), the Swiss Agency for Development and Cooperation (SDC), the Australian Centre for International Agricultural Research (ACIAR), the Norwegian Agency for Development Cooperation (Norad), the Federal Democratic Republic of Ethiopia, and the Government of the Republic of Kenya. The views expressed herein do not necessarily reflect the official opinion of the donors.

We express our gratitude to Dr. Segenet Kelemu for the insightful discussions; Mr. John Ngiela and Mr. Joseck Otiwi for their valuable support during sample collection; and Mr. Onesmus Wanyama, Mr. John Mark Makwatta, and Mr. James Kabii for their guidance on chemical and molecular biology instrumentation.

V.O.O. designed the study, collected and analyzed the data, and wrote the manuscript, M.N.G. conceptualized and designed the study, analyzed the data, wrote the manuscript, and mobilized the resources. C.K., S.M., and V.N.O. contributed to the bioinformatics data analysis part of the work. G.O.B. and J.M.O. supervised, reviewed, and edited the manuscript.

## AUTHOR AFFILIATIONS

[1]Animal Health Theme and Behavioural and Chemical Ecology Unit, International Centre of Insect Physiology and Ecology (*icipe*), Nairobi, Kenya
[2]Department of Chemistry, University of Nairobi (U.o.N), Nairobi, Kenya

## AUTHOR ORCIDs

Victor O. Omondi  http://orcid.org/0000-0003-0926-9789
Geoffrey O. Bosire  http://orcid.org/0000-0002-7289-6550
John M. Onyari  http://orcid.org/0000-0002-7289-6550
Samuel Mwasya  http://orcid.org/0009-0006-8629-4215
Vanessa N. Onyonyi  http://orcid.org/0009-0008-9978-2570
Merid N. Getahun  http://orcid.org/0000-0001-5902-9236

## FUNDING

| Funder | Grant(s) | Author(s) |
|---|---|---|
| Max Planck Institute for Chemical Ecology | B1126A-185 | Merid N. Getahun |
| International Centre of Insect Physiology and Ecology (ICIPE) | B1126A-185 | Merid N. Getahun |

## AUTHOR CONTRIBUTIONS

Victor O. Omondi, Data curation, Formal analysis, Investigation, Methodology, Writing – original draft | Geoffrey O. Bosire, Supervision, Writing – review and editing | John M. Onyari, Supervision, Writing – review and editing | Caleb Kibet, Data curation, Formal analysis | Samuel Mwasya, Formal analysis | Vanessa N. Onyonyi, Formal analysis | Merid N. Getahun, Conceptualization, Data curation, Formal analysis, Funding acquisition, Investigation, Methodology, Project administration, Writing – review and editing

## DATA AVAILABILITY

The data sets generated and/or analyzed during this study are all included in the paper and as supplemental materials. Additionally, all the analysis R scripts and packages used in this study are available at https://github.com/SamuelMwasya/Metataxonomics-rumen-microbes-visualization.

## ADDITIONAL FILES

The following material is available online.

### Supplemental Material

**Supplemental material (mSystems01228-23-S0001.docx).** Supplemental figure and tables.

### Open Peer Review

**PEER REVIEW HISTORY (review-history.pdf).** An accounting of the reviewer comments and feedback.

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
