## [Reviewer comments · mSystems]

Multi-omics analyses reveal rumen microbes and secondary metabolites that are unique to livestock species.

Victor Omondi, Geoffrey Bosire, John Onyari, Caleb Kibet, Samuel Mwasya, Vanessa Onyonyi, and Merid Getahun

Corresponding Author(s): Merid Getahun, International Centre of Insect Physiology and Ecology

Review Timeline:

Submission Date:	November 21, 2023
Editorial Decision:	December 11, 2023
Revision Received:	December 21, 2023
Accepted:	December 21, 2023

Editor: James Brown

Reviewer(s): Disclosure of reviewer identity is with reference to reviewer comments included in decision letter(s). The following individuals involved in review of your submission have agreed to reveal their identity: Richard Costa Polveiro (Reviewer #2)

Transaction Report:

DOI: <https://doi.org/10.1128/msystems.01228-23>

Re: mSystems01228-23 (Multi-omics analyses reveal rumen microbes and secondary metabolites that are unique to livestock species.)

Dear Dr. Merid N. Getahun:

Revision Guidelines

Sincerely,
James Brown
Editor
mSystems

Reviewer #2 (Comments for the Author):

The article, "Multi-omics analyses reveal rumen microbes and secondary metabolites that are unique to livestock species.", in Research Article format, the article carried out the metataxonomic characterization of microorganisms in the ruminal fluid of ruminants: (*Bos indicus*), goats (*Capra aegagrus hircus*), sheep (*Ovis aries*) and camels (*Camelus dromedaries*) from slaughterhouses in Nairobi and Machakos County. In addition, it carried out the identification and classification based on the

organic material in the rumen of possible plants related to the foods that the animals consumed, with PCR amplification, and extracted the metabolites from the rumen liquids of the species mentioned above and studied. The differences found in microbiota, the distinction between alpha and beta diversity, differences in the Venn graph, and differences found in diet and metabolites are very interesting.

The article showed significant improvement in writing and data presentation. The sections improve in their presentation, as do the figures. However, some minor suggestions are being made to the authors.

TITLE

OK.

ABSTRACT.

OK.

KEYWORDS

Line 63: Change words metagenomics: The correct word is Metataxonomic.

INTRODUCTION AND OBJECTIVE

OK.

MATERIALS AND METHODS

OK.

RESULTS.

Lines 155 - 166: This paragraph is very confusing. It is not possible to graphically differentiate which species of animal ruminant has greater or lesser diversity. Where did this information come from? Diversity and richness are different measures. Do not mix the terms cattle and cows. There are dots on the graphs with different colors (Fig. 2G) I highly recommend reviewing this.

Line 174 - 175: "Microbial diversity and secondary metabolites may be affected by host diet composition beside host's individual genetic makeup (20)". Great information, but it is placed out of place in the article. Submit the methodology, introduction, or discussion.

Lines 216-223: I reiterate that Figure 6 is interesting but not essential to the main body of the article. Send for supplementary material.

DISCUSSION

OK.

CONCLUSION

OK.

OTHERS

Lines 870-875: Figure Legends in Figure 1. Rewrite. It's difficult to understand the description.

Dear Dr James Brown, Editor in msystems, we thank you and the reviewer for the positive comments we received and that contributed for better clarification of our manuscript. We have addressed all yours and reviewer's comments point by point as outlined below

Reviewer #2 (Comments for the Author):

KEYWORDS

Line 63: Change words metagenomics: The correct word is Metataxonomic.

Author response: Thank you for the correction. We have changed the word metagenomics to Metataxonomic as suggested. Kindly see key words, pg. 3 line 63

RESULTS

Lines 155 - 166: This paragraph is very confusing. It is not possible to graphically differentiate which species of animal ruminant has greater or lesser diversity. Where did this information come from? Diversity and richness are different measures. Do not mix the terms cattle and cows. There are dots on the graphs with different colors (Fig. 2G). I highly recommend reviewing this.

Author response: Thank you for the insightful comment and correction, we have reanalyzed and made a new graph, we have excluded evenness from the figure as the reviewer pointed out it is already included in the Shannon index, that consider both abundance and evenness. We have rewritten the paragraph, as reviewer pointed out the variation was not clearly captured. Please see page 7 Line153-169

Additionally, the word cow has been replaced with cattle throughout the manuscript to avoid the mix up. We have also corrected the color mix up. Please see line 151-167, pg. 7 and fig 2

Line 174 - 175: "Microbial diversity and secondary metabolites may be affected by host diet composition beside host's individual genetic makeup (20)". Great information, but it is placed out of place in the article. Submit the methodology, introduction, or discussion.

Author response: Thank you for the comment, we have added justification why we did that to improve the connection or flow as suggested by the reviewer. It now reads 'Beside host's

individual genetic make-up, the composition of the host's diet influences the types and amounts of substrates available to the microbial community, which, in turn, influences the production of secondary metabolites (20). Therefore, we evaluated the dietary composition in the rumens of the four livestock species' Please see line 170-173, pg. 7.

Lines 216-223: I reiterate that Figure 6 is interesting but not essential to the main body of the article. Send supplementary material.

Author response: Thank you for the comment and suggestion. We have moved figure 6 to supplementary material as recommended. Please see supplementary material Fig S1.

OTHERS

Lines 870-875: Figure Legends in Figure 1. Rewrite. It's difficult to understand the description.

Author response: Thank you for the comment. We have re-written the Fig.1 legend. It now reads "Fig.1 Dominant bacterial and fungi across the four livestock species. (A) Bubble plots showing the qualitative and quantitative difference of unique and shared bacteria. (B) Venn diagram showing the total number of the identified unique and shared bacterial microbes. (C) Bubble plots showing the qualitative and quantitative difference of unique and shared fungi communities among the four livestock. (D) Venn diagrams show the total number of the identified unique and shared fungi. (E) Association analysis between selected microbes and host traits. Trait analysis done based on (13, 31-32, 35, 57, 83-93). Plots generated using microbes' relative abundance data with at least 1% relative abundance in one of the four livestock species. Please see pg. 31
Line 853-860

Re: mSystems01228-23R1 (Multi-omics analyses reveal rumen microbes and secondary metabolites that are unique to livestock species.)

Dear Dr. Merid N. Getahun:

Your manuscript has been accepted, and I am forwarding it to the ASM production staff for publication. Your paper will first be checked to make sure all elements meet the technical requirements. ASM staff will contact you if anything needs to be revised before copyediting and production can begin. Otherwise, you will be notified when your proofs are ready to be viewed.

Featured Image Submissions: If you would like to submit a potential Featured Image, please email a file and a short legend to mSystems@asmusa.org. Please note that we can only consider images that (i) the authors created or own and (ii) have not been previously published. By submitting, you agree that the image can be used under the same terms as the published article. File requirements: square dimensions (4" x 4"), 300 dpi resolution, RGB colorspace, TIF file format.

Sincerely,
James Brown
Editor
mSystems